# Glucose and Fructose Supplementation and Their Acute Effects on Anaerobic Endurance and Resistance Exercise Performance in Healthy Individuals: A Double-Blind Randomized Placebo-Controlled Crossover Trial

**DOI:** 10.3390/nu14235128

**Published:** 2022-12-02

**Authors:** Max L. Eckstein, Maximilian P. Erlmann, Felix Aberer, Sandra Haupt, Paul Zimmermann, Nadine B. Wachsmuth, Janis Schierbauer, Rebecca T. Zimmer, Daniel Herz, Barbara Obermayer-Pietsch, Othmar Moser

**Affiliations:** 1BaySpo—Bayreuth Center of Sport Science, Division of Exercise Physiology and Metabolism, University of Bayreuth, 95440 Bayreuth, Germany; 2Department of Endocrinology and Diabetology, Medical University of Graz, 8036 Graz, Austria; 3Department of Cardiology, Klinikum Bamberg, 96049 Bamberg, Germany; 4Department of Internal Medicine, Division of Endocrinology and Diabetology, Endocrinology Lab Platform, Medical University of Graz, 8036 Graz, Austria; 5Cardiovascular Diabetology Research Group, Division of Endocrinology and Diabetology, Department of Internal Medicine, Medical University of Graz, 8036 Graz, Austria

**Keywords:** glucose, lactate metabolism, resistance exercise, cycling, cardio-pulmonary exercise testing, fructose

## Abstract

Background: The effects of glucose, fructose and a combination of these on physical performance have been subject of investigation, resulting in diverse findings. Objective: The aim of this study was to investigate how an individualized amount of glucose, fructose, and a combination of these compared to placebo (sucralose) alter endurance performance on a cycle ergometer, lower and upper body resistance exercise performance at individualized thresholds in healthy young individuals. Methods: A total of 16 healthy adults (9 females) with an age of 23.8 ± 1.6 years and a BMI of 22.6 ± 1.8 kg/m^2^ (body mass (BM) 70.9 ± 10.8 kg, height 1.76 ± 0.08 m) participated in this study. During the screening visit, the lactate turn point 2 (LTP2) was defined and the weights for chest-press and leg-press were determined. Furthermore, 30 min prior to each exercise session, participants received either 1 g/kg BM of glucose (Glu), 1 g/kg BM of fructose (Fru), 0.5 g/kg BM of glucose/fructose (GluFru) (each), or 0.2 g sucralose (placebo), respectively, which were dissolved in 300 mL of water. All exercises were performed until volitional exhaustion. Time until exhaustion (TTE) and cardio-pulmonary variables were determined for all cycling visits; during resistance exercise, repetitions until muscular failure were counted and time was measured. During all visits, capillary blood glucose and blood lactate concentrations as well as venous insulin levels were measured. Results: TTE in cycling was 449 ± 163 s (s) (Glu), 443 ± 156 s (Fru), 429 ± 160 s (GluFru) and 466 ± 162 s (Pla) (*p* = 0.48). TTE during chest-press sessions was 180 ± 95 s (Glu), 180 ± 92 s (Fru), 172 ± 78 s (GluFru) and 162 ± 66 s (Pla) (*p* = 0.25), respectively. Conclusions: Pre-exercise supplementation of Glu, Fru and a combination of these did not have an ergogenic effect on high-intensity anaerobic endurance performance and on upper and lower body moderate resistance exercise in comparison to placebo.

## 1. Introduction

Athletes at any stage of professionality are trying to enhance their performance by proper sleep, recovery schemes and nutritional interventions. Especially for nutrition, several supplementation strategies for macro- and micro-nutrients have been investigated and optimized for different types of exercise, training and competition and are endorsed by different position statements [1,2]. While substances such as creatine, caffeine or beta-alanine have been investigated and proven as mostly ergogenic when consumed prior to exercise [3,4,5], the consumption of ‘pre-workout’ drinks, including a mixture of the aforementioned substances combined with different types of carbohydrates (CHO) to enhance individual performance prior to exercise activities, have been a matter of debate for decades [1,6,7]. CHO especially serve as important fuel source before and during exercise, which has been repeatedly demonstrated across various sports, e.g., swimming or running [8,9]. In beverages intending to improve exercise fitness, CHO serve as the main source of energy to enhance performance by maintaining euglycemia and altered fuel utilization [10]. Numerous studies have investigated the beneficial effects of glucose prior to and during exercise to increase, prolong and improve performance during exercise endurance tests such as in swimming or cycling [11,12]. Glucose has also shown to ameliorate the number of repetitions and weight being moved during resistance training [13]. Over recent decades, exercise scientists experimented with different combinations and amounts of glucose to match the demands of different types of exercise. Fructose has drawn attention due to its altered abilities to be metabolized in comparison to glucose. At rest, glucose demands insulin to enter the cell via glucose transporter type 4 (Glut-4) which may act independently of insulin during vigorous intensity exercise [14]. Fructose can enter the cell independently of insulin via Glut-2 and Glut-5 [15]. During that process, lactate is produced that may slightly elevate basal blood lactate levels, though this does not contribute to detrimental effects following subsequent exercise sessions [16]. However, supraphysiologic doses of glucose prior to exercise may be contraindicated since it may induce hyperglycemia that may potentially lead to hyperinsulinemia that consequently leads to hypoglycemia and discomfort of the individual if glycogen stores are not full [17].

Consequently, researchers have tried to overcome these obstacles by aiming to investigate a combination of glucose and fructose and their effects on performance [18,19]. Following these investigations, it was suggested that a combination of both substances may be preferred over glucose alone, particularly during moderate- to high-intensity exercise sessions [20]. However, the amounts that were administered in these studies were not body mass-individualized and may therefore be unspecific and not reproducible for other cohorts. Moreover, an individualized amount of CHO given prior to physical exercise would also be preferable since the amount of glucose uptake in the intestinal system is limited by sodium glucose transporters 1 (SGLT-1) while fructose is mainly limited by Glut-5, which would potentially make a body mass-individualized amount of substance given prior to physical exercise preferable. Due to the diverse existing data on this topic and the preliminary study datawe have conducted, we hypothesize that a combination of glucose and fructose may lead to a better performance in comparison to glucose, fructose and a placebo [17].

Considering these previous studies, the aim of this double-blind randomized placebo-controlled crossover trial was to investigate how an individualized amount of glucose, fructose, as well as a combination thereof compared to placebo influence on endurance performance on a cycle ergometer, lower body resistance exercise and upper body resistance exercise at individualized thresholds in healthy young individuals.

## 2. Materials and Methods

This was a single center, double-blind randomized, placebo-controlled crossover trial, assessing the impact of glucose, fructose, a combination thereof compared to placebo on physical exercise performance of healthy individuals. The local ethics committee of the University of Bayreuth (Germany) approved the study protocol (O 1305/1.GB, 15th November 2021), which was registered at the German Clinical Trials Register (DRKS00027153). The study was conducted in conformity with the declaration of Helsinki and Good Clinical Practice. Before any trial related activities, participants were informed about the study protocol and gave their written informed consent. The study was conducted between November 2021 and ended in April 2022. Overall, 224 trial visits were conducted.

### 2.1. Eligibility Criteria

Eligibility criteria included male or female individuals aged 18–65 years with a body mass index (BMI) of 18.0–29.9 kg/m^2^, both inclusive. Participants had to be metabolically healthy and normoglycemic after an overnight fast.

Individuals were excluded if they were participating in other ongoing studies or require investigational medicinal products. Systolic and diastolic blood pressure after resting for five minutes in a supine position had to be within the range of 90–150 and 50–95 mmHg, respectively. Participants were also excluded if they had a history of multiple and/or severe allergies or intolerances to any trial related products. Moreover, the regular or irregular intake of any medication with a potential impact on the parameters assessed (blood pressure lowering therapy, antiarrhythmic drugs, antidepressants with QT prolonging potential) was classified as exclusion criterion. An assessment via a medical investigator of all inclusion and exclusion criteria occurred at the screening visit before enrolment in the study. The inclusion and exclusion criteria can be found in the Appendix A.

### 2.2. Study Design

After being enrolled in the study, participants were assigned a participant code in an ascending order. This number operated as study ID throughout the course of the study. They were then allocated to the order in which the trial visits were conducted in a randomized crossover fashion set using the software Research Randomizer^®^ [21]. A researcher not otherwise involved in the trial randomized the order of exercises (1:1:1) and the order of consumed beverages in a crossover fashion (1:1:1:1) as shown in Figure 1.

At the start of each study visit, participants received either 1 g/kg body mass (BM) of glucose (Glu) (Fisher Scientific, Lough-borough, United Kingdom), 1 g/kg BM of fructose (Fru) (Grüssing, Felsum, Germany), and 0.5 g/kg BM of a mixture of glucose and fructose (GluFru) (each) dissolved in 300 mL water. The amount of placebo, Sucralose (MyProtein, Norwich, United Kingdom), was administered as a fixed 0.2 g per dosage with the intention to mimic the taste of the other trial products. Between each visit, a minimum period of 48 h was maintained to ensure that glycogen stores were replenished. Participants were allowed to eat regularly in this period but were told to avoid strenuous exercise sessions. This procedure was conducted for any of the twelve visits (four exercise sessions consisting of including cycling, chest- and leg-press visits).

### 2.3. Screening Visit

At the screening visit, participants were informed about all study-related procedures, given instructions and were asked to provide written informed consent. Medical staff requested and documented their medical history. Afterwards, the general health status of the participants was examined: body composition was assessed via bioelectrical impedance analysis (Inbody 720, Inbody Co., Seoul, Korea); body height was measured manually (Seca 217, Seca, Hamburg, Germany).

Furthermore a 12-lead ECG (CardioPart 12, Amedtec, Aue-Bad Schlema, Germany) was recorded with the participants resting in a supine position for 5 min, prior to which participants were instructed to remain in a lying position for at least 5 min. During this test, blood pressure measurements were also performed manually.

In addition, a capillary blood sample was collected from a hyperemized earlobe to quantify blood glucose concentration and analyzed for any form of dysglycemia in healthy adults (Biosen S-Line, EKF-Diagnostic, Barleben, Germany).

Participants also were asked about their physical activity via an international physical activity questionnaire in short form (IPAQ-SF). To familiarize themselves with the resistance exercises and being instructed about the setting of the chest- and leg-press, participants had to demonstrate at least five repetitions of the weight, that was moved according to their body during their prospective sessions of the study.

Subsequently, a cardio-pulmonary exercise (CPX) test was performed to assess the maximum power output and oxygen uptake. Heart rate was measured continuously via 12-lead ECG for safety reasons. Moreover, breath-by-breath measurements were conducted and averaged over 5 s for later analysis (METALYZER^®^ 3B; Cortex Biophysik GmbH, Leipzig, Germany). Blood glucose and lactate values were collected via capillary measurements from the earlobe at rest, after a 3-min warm-up phase, every minute, at the end and after cool down and quantified afterwards (Biosen S-Line, EKF-Diagnostic, Barleben, Germany).

At the beginning of the incremental CPX test, participants sat still on the cycle ergometer for 3 min without pedaling before starting the warm-up period of 3 min at a workload of 20 watts (W). After that, the mechanical power output was increased by 15 W/min (female) or 20 W/min (male) until exhaustion. A 3-min active recovery was conducted at 20 W followed by a 3-min passive recovery without pedaling. This test enabled to detect first and second lactate turn point 1 (LTP1) and lactate turn point 2 (LTP2) and recorded the maximum power output (P_max_) [22]. These parameters were relevant to determine the exercise intensity for the upcoming four cycling exercise sessions [22]. Participants subsequently tested resistance exercise devices if weights could be moved on leg-press (4100+, Cybex, Daytona, FL, USA) and chest-press (VR2, Cybex, Daytona, FL, USA).

### 2.4. Study Visits

Participants were asked to visit the laboratory in the morning after an overnight fast for at least 12 h which allowed only non-alcoholic or caffeinated beverages to be ingested. A gap of 48 h was kept between visits to ensure that glycogen stores were replenished. All participants were asked about any COVID-19 related symptoms prior to entering the laboratory environment and if the participant felt uncomfortable or deemed sick by the study team, the participant was sent home and the visit was rescheduled. The procedure of each visit started with measuring the body mass to ensure no significant increases or decreases occurred between visits.

Participants then received an opaque shaker bottle filled with a solution of 300 mL water and their individual amount of either Glu, Fru, GluFru or placebo. The shaker bottles were labeled with the study ID to avoid any distribution errors. The beverages and opaque shaker bottles were prepared by a researcher not otherwise involved in the trial. Drinks were handed out to the participants directly from the study team. Subsequently a cannula was placed in the antecubital vein for venous blood sampling (4 mL each) for insulin measurements. After that, participants were asked to drink the beverage as quickly as possible. The drinking time was defined as *timepoint −30,* the starting point of the exercise exactly 30 min later was defined as *timepoint 0*. For the next 24 min (cycling exercise) and 27 min (strength exercises), participants had to rest in an upright and resting position. During the ergometer sessions, participants were equipped with a 12-lead ECG for the detection of heart rate curves and a spirometric device for breath-by-breath analysis (METALYZER^®^ 3B, Cortex Biophysik GmbH, Leipzig, Germany). The exercise test on the cycle ergometer was initiated with a 3-min resting period (24 min after drinking) followed by a 3-min warm-up at 20 W. Strength exercises were performed without a 12-lead ECG and spirometric device. Participants performed the 3-min warm-up at 20 W before strength exercises similar to the cycling sessions.

Similar positions during each exercise were ensured, ergometer and resistance exercise settings were documented at the screening visit and fitted from the study team before each exercise. Moreover, load (100% of body mass for leg-press; 30% (female) / 75% (male) BM for chest-press) and power output (20% watts above LTP2) for the participant, that were assessed during the screening visit before, were set by the study team. These weights were chosen since these were considered movable and safe by the research team to be performed until volitional exhaustion in non-experienced resistance exercise trained individuals.

During the period of exercise, participants were verbally motivated to perform until volitional exhaustion (*timepoint END*). Volitional exhaustion was defined as the individual inability to perform a further repetition during strength exercises or to keep the cadence above 60 rpm on the cycling ergometer. After that, participants were asked to remain seated in an upright position on the cycle ergometer for 3 min. They were then allowed to dismount from the ergometer and asked to remain seated at the laboratory until 30 min after *timepoint END* to fulfill every measurement. In case of the strength exercises, participants could directly exit the leg or chest-press, but also had to stay at the laboratory for consecutive measurements for the next 30 min.

Venous blood samples were collected prior to exercise before drinking (30 min from starting the exercise = *timepoint −30*), at the start of the exercise (*timepoint 0*), at the end of exercise (*timepoint END*) and 15 (*timepoint +15*) and 30 min (*timepoint +30*) following exercise.

### 2.5. Blood Sampling

Over the study period, these blood samples were stored pseudonymized (study ID and visit number only) at −80 °C at the University of Bayreuth (Division of Exercise Physiology & Metabolism). Capillary blood samples were collected (glucose & lactate, 20 μL) from the earlobe at the same timepoints as venous blood samples (if applicable) and analyzed via enzymatic-amperometric method (Biosen S-line, EKF Diagnostics, Barleben, Germany). Additionally, during the course of cycling exercise capillary blood samples were also collected 6 min before the start, every completed minute from *timepoint 0* and right at the time of exhaustion (*timepoint END*) and 3 min after *timepoint END.* For aligned presentation, data are shown in 2 min intervals. Once the study was completed, plasma samples were analyzed by routine clinical biochemistry assays for insulin (Advia Centaur XPT, Siemens, Munich, Germany).

### 2.6. Statistical Analysis

Data were analyzed in GraphPad Prism 8.0.2 (GraphPad, LA Jolla, CA, USA) and tested for normal distribution via Shapiro-Wilk test. Throughout the presentation of the results of the study, data will be presented according to their distribution. For the primary and secondary outcome, data were compared via analysis of variance (ANOVA) for repeated measurements or fitting mixed models with Tukey’s post-hoc test, if required. Statistical significance was accepted at *p* < 0.05 (two-tailed). A sample size calculation based on previous findings by Baur et al. [18] using a paired *t*-test (two-sided, alpha 5%, power 90%) estimated that 16 participants were required to demonstrate statistical significance for the results of this study.

## 3. Results

The group of participants consisted of 16 healthy adults (9 females) with a mean ± SD age of 23.8 ± 1.6 years and a BMI of 22.6 ± 1.8 kg/m^2^ (BM 70.9 ± 10.8 kg, height 1.76 ± 0.08 m). Male participants had a relative VO_2max_ of 52 ± 8 mL/kg/min, a peak power of 334 ± 25 W and a maximum heart rate of 190 ± 8 bpm at the screening CPX test. Female participants had a relative VO_2max_ of 41 ± 5 mL/kg/min, a peak power of 236 ± 33 W and a maximum heart rate of 188 ± 8 bpm. One screened participant was not eligible to participate in the study because of a fructose hypersensitivity. No participant had to be withdrawn or left the study prematurely. Detailed study results can be seen in Figure 2.

### 3.1. Cycling

A total of 14 out of 16 participants were included for analysis. One individual had an episode of asymptomatic ventricular extrasystoles with a 4:1 ratio during the first study specific cycling test. This might trigger ventricular arrhythmias during the test as judged by the medical investigator and therefore he was excluded for further cycling exercises for safety reasons. The other individual did not perform the CPX test at the screening visit until volitional exhaustion that caused an incorrect LTP2 assessment for the following exercise tests. This led to an outlier that had to be excluded from the analysis.

In response to Glu, cycling duration was of 449 ± 163 s, Fru 443 ± 156 s, GluFru to 429 ± 160 s and placebo to 466 ± 162 s (*p* = 0.48).

In women, Glu led to a cycling duration 419 ± 100 s, Fru to 409 ± 161 s, GluFru to 393 ± 115 s and placebo to 446 ± 169 s (*p* = 0.51). In men, Glu led to a cycling duration of 506 ± 245 s, Fru 487 ± 150 s, GluFru to 493 ± 221 s and placebo to 502 ± 160 s (*p* = 0.69).

### 3.2. Chest-Press

All 16 participants completed all four chest-press sessions. Overall, Glu led to an exercise time of 180 ± 95 s, Fru to 180 ± 92 s, GluFru to 172 ± 78 and placebo to 162 ± 66 s (*p* = 0.25). In women, an exercise time of 240 ± 93 s, Fru to 239 ± 86 s, GluFru to 228 ± 65 s and placebo to 206 ± 59 s (*p* = 0.23). In men, Glu led to an exercise time of 112 ± 26 s, Fru to 113 ± 31 s, GluFru to 108 ± 23 s and placebo to 113 ± 25 s (*p* = 0.78). Regarding the overall number of repetitions, Glu led to 58 ± 31 repetitions, Fru 63 ± 28 repetitions, GluFru 57 ± 24 repetitions and placebo to 56 ± 23 repetitions (*p* = 0.20). In women, Glu led to 77 ± 31 repetitions, Fru to 82 ± 23 repetitions, Glufru 73 ± 17 repetitions and placebo to 71 ± 19 repetitions (*p* = 0.26) In men, Glu led to 35 ± 9 repetitions, Fru to 39 ± 10 repetitions, GluFru to 37 ± 11 repetitions and placebo to 37 ± 9 repetitions (*p* = 0.46).

### 3.3. Leg-Press

All 16 participants completed all visits for leg-press. Overall, Glu led to an exercise time of 339 ± 220 s, Fru to 385 ± 246 s, GluFru to 365 ± 242 and placebo to 384 ± 318 s (*p* = 0.41). In women, Glu led to an exercise time of of 391 ± 279 s, Fru 400 ± 294 s, GluFru to 419 ± 303 s and placebo 444 ± 411 s (*p* = 0.69). In men, Glu led to an exercise time of 282 ± 123 s, Fru to 369 ± 200 s, GluFru to 293 ± 115 s and placebo to 315 ± 170 s (*p* = 0.12). Regarding the number of repetitions overall Glu led to 112 ± 77, Fru to 127 ± 86, GluFru to 114 ± 79 and placebo to 126 ± 106 repetitions (*p* = 0.36). In women Glu led to 126 ± 96, Fru to 131 ± 102, GluFru to 126 ± 97 and placebo to 135 ± 128 repetitions (*p* = 0.83). In men Glu led 94 ± 41, Fru to 120 ± 67, GluFru to 97 ± 76 and placebo to 114 ± 75 repetitions (*p* = 0.22).

### 3.4. Venous Blood Samples

Results of venous blood samples (insulin) are shown in Figure 3.

### 3.5. Parameters of during Cycling Study Visits

No significant differences between Glu, Fru, GluFru and placebo were found in relative oxygen consumption, oxygen pulse and heart rate (*p* > 0.05). Respiratory exchange ratio showed significant differences at −3 min of fructose and GluFru compared to glucose and placebo (*p* < 0.05). At timepoint 0, only Fru (*p* < 0.0001) and GluFru (*p* = 0.02) were significantly higher compared to placebo. Results can be seen in more detail in Figure 4.

### 3.6. Physical Activity

No statistical differences were found between visits in regards of physical activity measured via IPAQ. For cycling visits Glu had a total of 3609 ± 1121, Fru 3597 ± 1282, GluFru 3904 ± 1247 and placebo of 4018 ± 1190 MET-min/week (*p* = 0.60)

During the chest-press visits had a total MET-min/week of 3340 ± 1383 for Glu, 3577 ± 2261 for Fru, 3428 ± 1538 for GluFru and 3226 ± 1196 for placebo (*p* = 0.77). At leg-press visits Glu led to 3950 ± 1851, Fru to 3951 ± 1338, GluFru to 4038 ± 1885 and placebo to 4114 ± 1565 Met-min/week (*p* = 0.96).

### 3.7. Other

All female participants in this study were taking contraceptives during the course of the study. At the onset of their period, the performance of the exercise tests was not significantly different when compared to other exercise visits. During CPX, individuals at onset of their period had performed for 382 ± 133 s compared to 367 ± 129 s (*p* = 0.22). During leg-press sessions, individuals at onset of their period performed 104 ± 56 repetitions compared to 125 ± 90 repetitions (*p* = 0.12). A comparison for chest-press could not be conducted due to lack of pairs for statistical comparison.

## 4. Discussion

This is the largest study comprehensively assessing the effects of pre-exercise Glu, Fru and GluFru compared to placebo in different endurance and resistance exercises in a trained cohort of male and female healthy individuals. This study has shown that pre-exercise CHO independent of their composition of Glu and Fru had no impact on performance in comparison to the control arm including sucralose. Research from Choi et al. indicated a lactate-induced insulin resistance and impaired insulin signaling, leading to a decreased insulin-stimulated glucose transport in the skeletal muscle that is in line with our results [23].

Our findings deliver a lot of information that should be carefully considered by athletes, coaches and real-world practitioners. First and foremost, the results of our study are not entirely new, since previous studies found likely benefits of CHO supplementations in comparison to placebo during cycling [18,24]. Others omitted the effect of CHO on performance [19] and reviews cautiously recommended GluFru for moderate- to high-intensity exercise sessions [20]. Considering this, our study demonstrates clear evidence that pre-exercise CHO consumption, independent of composition, does not have an ergogenic effect on performance in high-intensity anaerobic endurance exercise and moderate resistance exercise. The metabolic responses at rest seen in this study following the consumption of CHO have previously been investigated by our research group [17,25,26]. On this basis, this study was conducted since the time until reaching peak blood glucose and peak lactate values are reached in healthy individuals roughly 30 min after consumption which could have had the implication of a higher performance in comparison to placebo. Previous studies conducting similar exercise tests on a cycle ergometer at roughly 85% of VO_2max_ showed significant improvements in performance in half the sample size and solely male participants which cannot be considered as representative [27]. It might be considered that pre-exercise CHO consumption may lead to altered results between genders, since placebo was favored in women while in men it led to the second best outcome with marginal differences to glucose during cycling. A previous review has highlighted an altered CHO response in women compared to men [28]. Women have shown a significantly decreased CHO oxidation in response to glucose [29]. Women generally oxidize less CHO and more lipids due to a higher presence of estrogen [30]. Female contraceptive use is associated with low estrogen levels and a reduction in insulin levels which all female participants in our study were consuming that could have had an impact on our findings [31].

Even though CHO supplementation does not lead to ergogenic effects on performance during high-intensity anaerobic endurance exercise and moderate resistance exercise, one should carefully consider that gender specific differences in response to glucose and fructose are evident and this may demand future investigation. From this aspect our study is not without limitations. Since the included participants were young and healthy, yet without a specific training background of the tested types of exercise, it would be of interest how CHO supplementation would impact performance in well-trained cyclists or in well-resistance trained individuals. Due to the low amount of weight, the measured standard deviations were quite high and led to a large inter-individual variability that could have influenced levels of statistical significance in our study. In addition, we did not hand out food diaries over the entire course of the study since participants were asked to arrive fasted prior to each visit to avoid any kind of bias. Participants were encouraged to replenish glycogen storage after each visit which was ensured via a 48-h break between visits. However, for safety reasons and the level of experience of the participants, we were unable to increase the weights. Since the metabolic responses to the applied protocol at rest are well-researched, our findings may deliver valuable information for male and female athletes consuming CHO drinks regularly prior to exercise [17,25,26].

## 5. Conclusions

Glu, Fru and a combination of these do not enhance performance in comparison to placebo. We found gender specific differences that may demand future research investigating the impact of CHO supplementation in our trained study population. Until then, summarizing our results, there is no need to supplement CHO prior to physical activity with the aim of enhancing performance as long as rest periods have been complied and glycogen stores are replenished.

## Figures and Tables

**Figure 1 nutrients-14-05128-f001:**
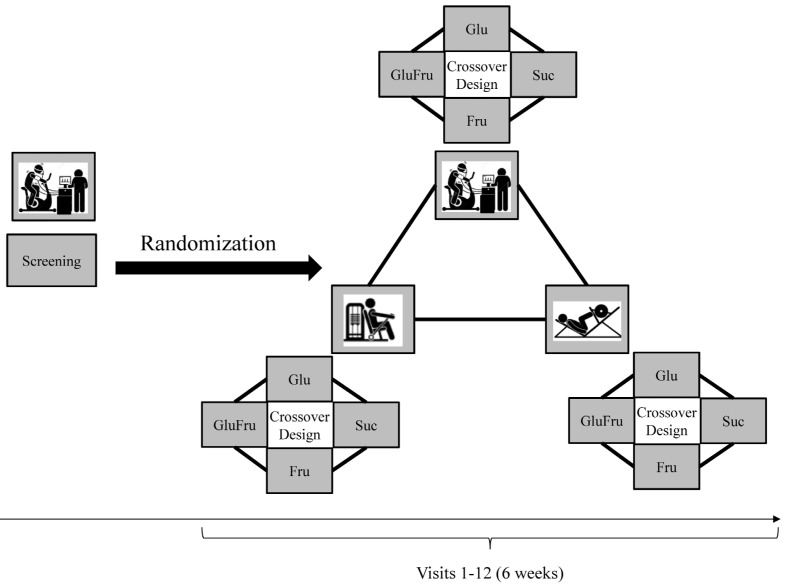
Study flow chart. Glu: Glucose, Fru: Fructose, GluFru: Glucose and fructose and Suc: Sucralose (placebo).

**Figure 2 nutrients-14-05128-f002:**
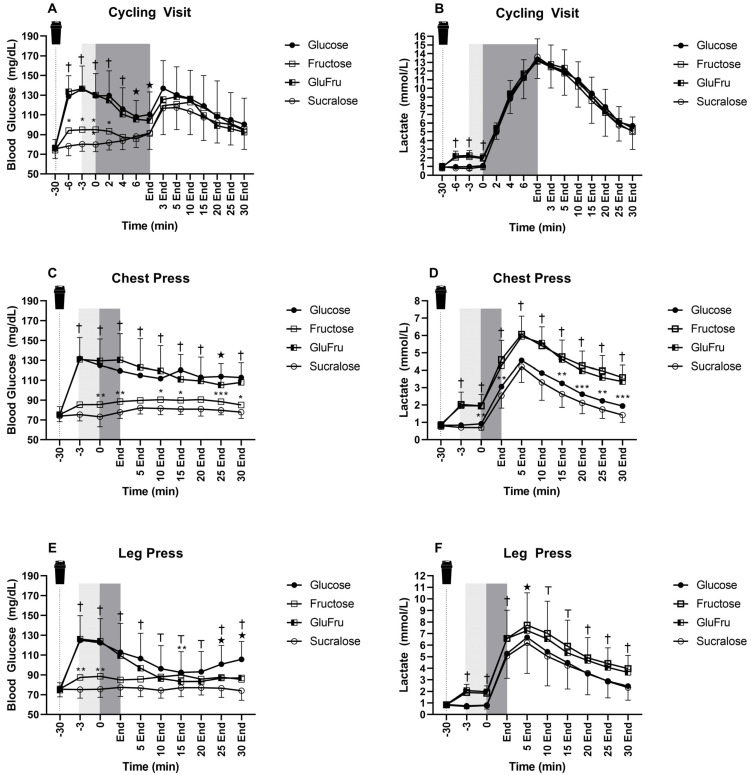
(**A**) Blood lactate during cycling study visits. Black circles indicate Glu. Open Squares indicate fru. Half open squares indicate GluFru. Open circles indicate sucralose. Light grey areas indicate warm up periods. Dark grey areas indicate exercise testing. † indicates statistical significance of Glu and GluFru compared to fructose and sucralose (*p* < 0.001). * indicates statistical significance between fructose and sucralose (*p* < 0.05), Stars indicate statistical significance of glucose compared to fructose and sucralose (*p* < 0.01). (**B**) Blood lactate during cycling study visits. † indicates statistical significance of Fru and GluFru compared to glucose and sucralose (*p* < 0.001). (**C**) Blood glucose during chest-press visits. † indicates statistical significance of Glucose and GluFru compared to Fructose and sucralose (*p* < 0.001). * indicates statistical significance between fructose and sucralose (*p* < 0.05), ** indicates statistical significance between fructose and sucralose (*p* < 0.01), *** indicates statistical significance between fructose and sucralose (*p* < 0.001). Stars indicate statistical significance of glucose compared to Glufru, fructose and sucralose (*p* < 0.05). (**D**) Blood lactate during chest-press visits. † indicates statistical significance of Fructose and GluFru compared to glucose and sucralose (*p* < 0.001). ** indicates statistical significance between glucose and sucralose (*p* < 0.01), *** indicates statistical significance between glucose and sucralose (*p* < 0.001). (**E**) Blood glucose during leg-press visits. † indicates statistical significance of glucose and GluFru compared to fructose and sucralose (*p* < 0.001). T indicates statistical significance of glucose compared to fructose, GluFru and sucralose (*p* < 0.01). ** indicates statistical significance of fructose in comparison to sucralose (*p* < 0.01). Stars indicate statistical significance of glucose compared to fructose, GluFru and sucralose. While fructose and GluFru are statistically different to sucralose (*p* < 0.05). (**F**) Blood lactate during leg-press visits. † indicates statistical significance of fructose and GluFru compared to glucose and sucralose (*p* < 0.001). T indicates statistical significance of glucose compared to fructose, fructose and glufru compared to sucralose (*p* < 0.05). Stars indicate statistical significance between fructose and sucralose (*p* < 0.05).

**Figure 3 nutrients-14-05128-f003:**
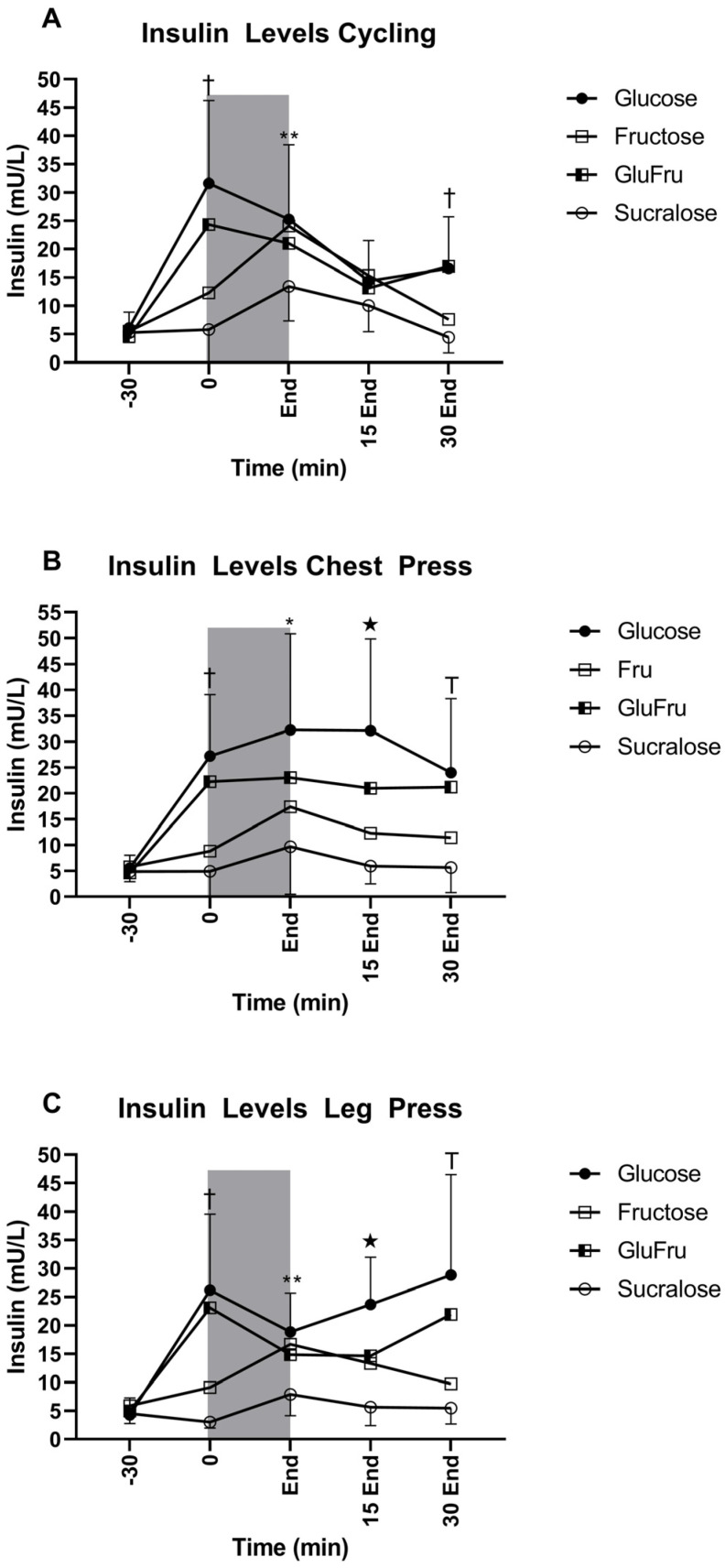
Black circles indicate Glu. Open Squares indicate fru. Half open squares indicate GluFru. Open circles indicate sucralose. Dark grey areas indicate exercise testing. (**A**) Insulin levels during cycling study visits † indicates statistical significance of glucose and GluFru compared to Fructose and sucralose (*p* < 0.01). ** indicates statistical significance of Glu, Fru, and GluFru compared to sucralose (*p* < 0.01). (**B**) † indicates statistical significance of glucose and GluFru compared to Fructose and sucralose (*p* < 0.01). * indicates statistical significance of Glu, Fru, and GluFru compared to sucralose (*p* < 0.01). Glu is significantly different to Fru (*p* < 0.05). Stars indicate statistical significance between all substances (*p* < 0.01). T indicates statistical difference between all substances besides Glu and Glufru (*p* < 0.01). (**C**) † indicates statistical significance of glucose and GluFru compared to Fructose and sucralose (*p* < 0.01). ** indicates statistical significance of Glu, Fru and GluFru compared to sucralose with no significant difference between glucose and Glufru and Fru and sucralose (*p* < 0.01). Stars indicate statistical significance between all substances (*p* < 0.05). T indicates statistical significance between all substances besides glucose and GluFru (*p* < 0.05).

**Figure 4 nutrients-14-05128-f004:**
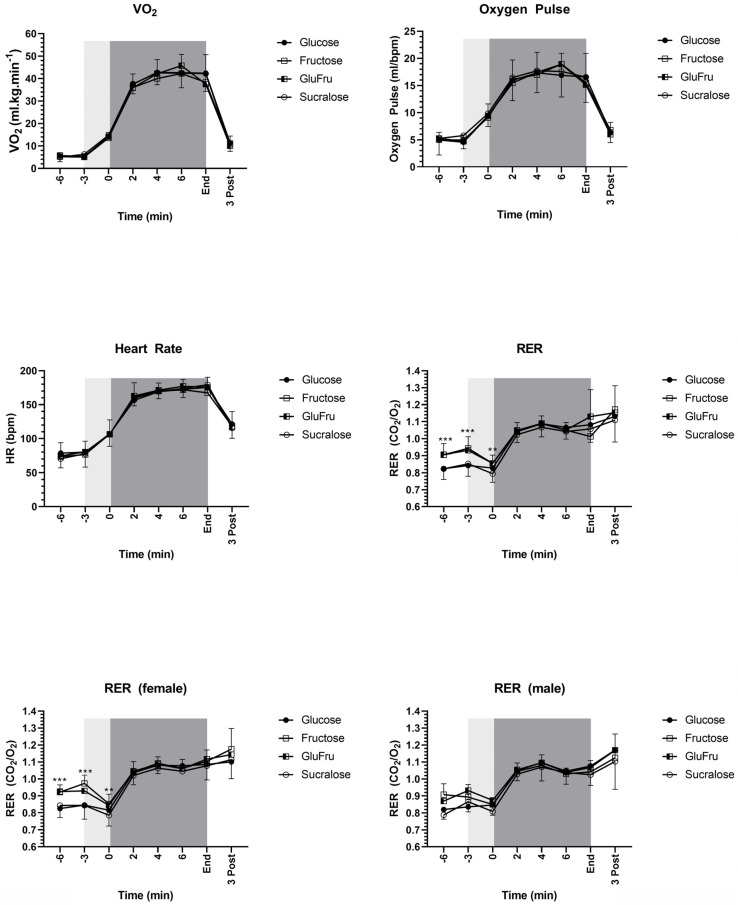
Black circles indicate Glu. Open Squares indicate fru. Half open squares indicate GluFru. Open circles indicate sucralose. Light grey areas indicate warm up periods. Dark grey areas indicate exercise testing. Parameters derived from cycling study visits. VO_2_: Oxygen consumption. CO_2_: Carbon dioxide production. RER: Respiratory exchange ratio. HR: Heart rate. *** indicates statistical significance between fructose and GluFru compared to glucose and placebo. ** indicates statistical significance between fru and Glufru compared to placebo.

## Data Availability

The data is available upon reasonable request.

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
