# Peer review of "Glucose and Fructose Supplementation and Their Acute Effects on Anaerobic Endurance and Resistance Exercise Performance in Healthy Individuals: A Double-Blind Randomized Placebo-Controlled Crossover Trial"

_nutrients, 2022, doi:10.3390/nu14235128_

Round 1

Reviewer 1 Report

This study is very great and has a good audience for readers of Nutrients. The study aimed to investigate how an individualized amount of glucose, fructose, a combination of these compared to placebo (sucralose) alter endurance performance on a cycle ergometer, lower and upper body resistance exercise performance at individualized thresholds in healthy young individuals.

-Aim: To change for: "This study to investigate how an individualized amount of glucose, fructose or combination of these compared to placebo (sucralose) alter endurance performance in healthy young individuals"

-Abstract: To reduce the size.

-Introduction: What is the hypothesis of study? Please, add also what is amount of carbohydrate oxidated per min.

-Methods: Is not clear. Whats the meaning "Individuals were excluded if they were participating in other ongoing studies or require investigational medicinal products". What are medicianal products? Carbohydrate, vitamins and protein supplements were allowed? In addition, what is habitual and pre test food intake? Please, to add it.

-Results: Dietary and supplements intake is crucial. 

9 females and 7 men is not adequate. This small sample size did not publication in Nutrients. Why mix gender?

-Discussion: The authors described "This study has shown that pre-exercise CHO independent of their composition of Glu and Fru had no impact on performance in comparison to the control arm including sucralose". However, no data regarding the previous food intake were reported. In addition, this information was not adressed as a limitation. 

-Conclusion: "well-trained population". This is not populational study. 

Author Response

Dear Editor, guest Editor

Dear Reviewer 1,2 & 3,

we would like to thank you for taking to review our manuscript and adding your valuable comments.

Reviewer 1

This study is very great and has a good audience for readers of Nutrients. The study aimed to investigate how an individualized amount of glucose, fructose, a combination of these compared to placebo (sucralose) alter endurance performance on a cycle ergometer, lower and upper body resistance exercise performance at individualized thresholds in healthy young individuals.

Thank you very much for your comment. We appreciate it a lot.

-Aim: To change for: "This study to investigate how an individualized amount of glucose, fructose or combination of these compared to placebo (sucralose) alter endurance performance in healthy young individuals"

Thank you for your comment. We amended it accordingly in the abstract

Page 1, Line 25-28

Objective: The aim of this study was to investigate how an individualized amount of glucose, fructose, a combination of these compared to placebo (sucralose) alter endurance performance on a cycle ergometer, lower and upper body resistance exercise performance at individualized thresholds in healthy young individuals.”

-Abstract: To reduce the size.

Thank you for your comment. We have reduced it.

Abstract:

Abstract: Background: Effects of glucose, fructose and combination of these on physical performance have been subject of investigation with diverse findings. Objective: The aim of this double-blind randomized placebo-controlled crossover study was to investigate how an individualized amount of glucose, fructose, a combination of these compared to placebo (sucralose) alter endurance performance on a cycle ergometer, lower and upper body resistance exercise performance at individualized thresholds in healthy young individuals. Methods: 16 healthy adults (9 females) with an age of 23.8 ± 1.6 years and a BMI of 22.6 ± 1.8 kg/m2 (body mass (BM) 70.9 ± 10.8 kg, height 1.76 ± 0.08 m) participated in this study. During the screening visit, the lactate turn point 2 (LTP2) was defined and the weights for chest-press and leg-press were determined. 30 min prior to each exercise session, participants received either 1 g/kg BM of glucose (Glu), 1 g/kg BM of fructose (Fru), 0.5 g/kg BM of glucose/fructose (GluFru) (each), or 0.2 g sucralose (placebo), respectively, which were dissolved in 300 mL of water. All exercises were performed until volitional exhaustion. Time until exhaustion (TTE) and cardio-pulmonary variables were determined for all cycling visits; during resistance exercise, repetitions until muscular failure were counted and time was measured. During all visits, capillary blood glucose and blood lactate concentrations as well as venous insulin levels were measured. Results: TTE in cycling was 449 ± 163 seconds (s) (Glu), 443 ± 156 s (Fru), 429 ± 160 s (GluFru) and 466 ± 162 s (Pla) (p=0.48). TTE during chest press sessions was 180 ± 95 s (Glu), 180 ± 92 s (Fru), 172 ± 78 s (GluFru) and 162 ± 66 s (Pla) (p=0.25), respectively. Conclusions: Pre-exercise supplementation of Glu, Fru and a combination of these did not have an ergogenic effect on high-intensity anaerobic endurance performance and on upper and lower body moderate resistance exercise in comparison to placebo.

-Introduction: What is the hypothesis of study? Please, add also what is amount of carbohydrate oxidated per min.

Thank you for your comment. We have refined the hypothesis of the study within the introduction and hope this is in agreement with the suggestion of the reviewer.

Page 3 Line 93 to Line 96

Due to the diverse existing data on this topic and the preliminary study data we have conducted we hypothesize that a combination of glucose and fructose may lead to a better performance in comparison to glucose, fructose and a placebo [1,2].

Considering these previous studies, the aim of this double-blind randomized placebo-controlled crossover trial was to investigate how an individualized amount of glucose, fructose, as well as a combination thereof compared to placebo influence endurance performance on a cycle ergometer, lower body resistance exercise and upper body resistance exercise at individualized thresholds in healthy young individuals. “

Regarding your second comment: It is rather unusual to add information of the results in the introduction. We have conducted a preliminary study already regarding the carbohydrate oxidation at rest with the exact same study setting at rest to strengthen our hypothesis in this study design. You can find the change in carbohydrate oxidation in Eckstein, M.L.; Brockfeld, A.; Haupt, S.; Schierbauer, J.R.; Zimmer, R.T.; Wachsmuth, N.; Zunner, B.; Zimmermann, P.; Obermayer-Pietsch, B.; Moser, O. Acute Metabolic Responses to Glucose and Fructose Supplementation in Healthy Individuals: A Double-Blind Randomized Crossover Placebo-Controlled Trial. Nutrients 2021, 13, 4095, doi:10.3390/nu13114095 at Table 1. We hope this answers your query sufficiently, since the addition of this data would not be adequate in the introduction of the manuscript.

-Methods: Is not clear. Whats the meaning "Individuals were excluded if they were participating in other ongoing studies or require investigational medicinal products". What are medicianal products? Carbohydrate, vitamins and protein supplements were allowed? In addition, what is habitual and pre test food intake? Please, to add it.

Thank you for your comment. This implements that participants that were involved in this study were not allowed to be enrolled in other studies were investigational medical products were tested/ or at which medical products were consumed in an off-label manner. We have included this to ensure that no bias due to other medication would occur that could influence our study results. None of our participants were consuming any type of supplementation (protein powders) or vitamins (tables/capsules) during the course of the study. In addition, participants had to show up fasted prior to each visit to ensure the starting set-up would remain similar across all study-related visits.
This is shown in:

Page 3: Line 112 – 114

“2.1. Eligibility criteria

Eligibility criteria included male or female individuals aged 18–65 years with a body mass index (BMI) of 18.0–29.9 kg/m2, both inclusive. Participants had to be metabolically healthy and normoglycemic after an overnight fast. “

Page 5 Line 185 – 193

“2.4. Study visits

Participants were asked to visit the laboratory in the morning after an overnight fast for at least 12 hours which allowed only non-alcoholic or caffeinated beverages to be ingested. A gap of 48 h was kept between visits to ensure that glycogen stores were replenished. All participants were asked about any COVID-19 related symptoms prior to entering the laboratory environment and if the participant felt uncomfortable or deemed sick by the study team the participant was sent home and the visit was rescheduled. The procedure of each visit started with measuring the body mass to ensure no significant increases or decreases occurred between visits. “

-Results: Dietary and supplements intake is crucial. 

Thank you for your comment, we agree. Due to the set-up of the study we were making sure, that participants had sufficient time to replenish glycogen stores between each visit. In addition, all participants had to be fasted prior to each exercise test. We achieved this goal since you can see in Figure 2 and Figure 3 that blood glucose levels were low (70 mg/dL) and insulin levels too, due to the overnight fasting prior to the consumption of the study beverage. If there had been any consumption of supplements or other dietary intake (and/or caffeinated drinks) there would have been a physiological response that we would have detected. If there was any violation of the study protocol measured by either insulin or blood glucose prior to each visits we would have sent home the participant for another visit. This has not happened in the course of the study hence we believe that the data gathered is valid.

9 females and 7 men is not adequate. This small sample size did not publication in Nutrients. Why mix gender?

Thank you very much for your comment. It is common to include both genders in a study to investigate the overall response and to separately determine the results (subgroup analysis). We are unsure what is meant by the comment of the reviewer. Since we have conducted 224 study visits (including screening visit) in a complex cross-over design to exclude any form of bias. In addition, we have conducted a sample size estimation which in comparison to other exercise studies investigating intra-participant exercise responses to a supplement is very precise and delivers valid statistical information. It is common to investigate both, men and women to represent the metabolic response for each gender in general and in the subgroup.

-Discussion: The authors described "This study has shown that pre-exercise CHO independent of their composition of Glu and Fru had no impact on performance in comparison to the control arm including sucralose". However, no data regarding the previous food intake were reported. In addition, this information was not adressed as a limitation. 

Thank you very much for your comment. We have conducted each study visit with a >12h fasting period. This means that participants were not allowed to consume any food or caloric dinks within this period. This is confirmed by our results. Participants were encouraged to replenish glycogen storage after each study visit until the next visit. A distinct food diary was not conducted only a physical activity measurement (IPAQ) was conducted to ensure regular physical activity was obeyed to and the study did not have a further impact on the overall life of the participant. However, we have addressed it as a limitation of the study.

Page 13, Line 422-426

“In addition, we did not hand out food diaries over the entire course of the study since participants were asked to arrive fasted prior to each visit to avoid any kind of bias. Participants were encouraged to replenish glycogen storage after each visit which was ensured via a 48h break between visits.”

-Conclusion: "well-trained population". This is not populational study. 

Thank you for your comment. We apologize for not defining it correctly. We have changed it into the following and hope this is in agreement with the reviewers’ suggestion

Page13 Line 432 to 434

“Glu, Fru and a combination of these do not enhance performance in comparison to placebo. We found gender specific differences that may demand future research investigating the impact of CHO supplementation in our trained study population.”

Reviewer 2 Report

Thank you that you give me opportunity to review this manuscript   Glucose and fructose supplementation and their acute effects on anaerobic endurance and resistance exercise performance in healthy individuals: a double-blind randomized placebo-con-4 trolled crossover trial.

The authors conducted and reported the interesting study, although it includes very few participants.

The study was planned and conducted in detail, however, in some places in the manuscript, the description needs to be clarified.

Some comments on the manuscript are described below:

1.      Title  Glucose and fructose supplementation and their acute effects on anaerobic endurance and resistance exercise performance in healthy individuals: a double-blind randomized placebo-con-4 trolled crossover trial.

Only a dozen or so people took part in the study, so I would add a preliminary study in the title. This will "protect" the statistical calculations, which with such a very small group, giving statistical significance when comparing two even smaller groups of participants is a bit of an exaggeration.

2.      Material and Methods

Line 103 “…..of healthy individuals”.  It's not quite true because on line 111 the authors wrote ….BMI 18.0-29.9 . This is overweight, almost obese (BMI is from 30 and above). And  in line 115….. the range of 90–150 and 50– 95 mmHg, and 150/90 mmHg is stage I hypertension. Please,, explain it.

Line 111- “Participants had to be metabolically healthy…  what do you mean the metabolically healthy, Please, explain it.

2.1. Eligibility criteria .   The criterion for inclusion or exclusion of participants from the study is not precisely described. Please, correct it.

2.2. Study design.

What was the sample selection?

When was the study conducted?

How many trial visits were there?

Where and how were participants selected for the study?

Did the participants receive any financial gratification?

Line 158 Participants also were asked about their physical activity via an international physical 158 activity questionnaire in short form (IPAQ-SF). There is no references, please add it.

Line 168 Blood glucose and lactate values were collected via capillary measurements  from the earlobe at rest, after a 3-minute warm-up phase, every minute, at the end and after cool down and quantified afterwards.  How many times?

      Line 188….the participant was sent home and the visit was rescheduled. How and when were they re-enrolled in the study?

3.      Results

Please add age range of participants.

Line 257-259 One screened participant was not eligible to participate in the study because of a fructose hypersensitivity. No participant had to be  withdrawn or left the study prematurely.

But in Line 261 14 out of 16 participants were included for analysis

And in Line 261-4 One individual had an episode  of asymptomatic ventricular extrasystoles with a 4:1 ratio during the first study specific cycling test. This might trigger ventricular arrhythmias during the test as judged by the medical investigator and therefore he was excluded for further cycling exercises for safety  reasons.

Please explain it, how many participants took part in this study.

4.      Discussion

In the discussion, the authors from line 358 to 380, almost one page, describe the results, this should be described in the 3rd results. This is not a discussion. Please correct it.

The weakest element of the study is too small a study population.

Author Response

Reviewer 2

Thank you that you give me opportunity to review this manuscript “   Glucose and fructose supplementation and their acute effects on anaerobic endurance and resistance exercise performance in healthy individuals: a double-blind randomized placebo-con-trolled crossover trial.

The authors conducted and reported the interesting study, although it includes very few participants.

The study was planned and conducted in detail, however, in some places in the manuscript, the description needs to be clarified.

Some comments on the manuscript are described below:

Thank you very much for your comments, reading our manuscript and investing your valuable time.

  1. Title “Glucose and fructose supplementation and their acute effects on anaerobic endurance and resistance exercise performance in healthy individuals: a double-blind randomized placebo-con-trolled crossover trial.

Only a dozen or so people took part in the study, so I would add a preliminary study in the title. This will "protect" the statistical calculations, which with such a very small group, giving statistical significance when comparing two even smaller groups of participants is a bit of an exaggeration.

Thank you for your comment. We do not agree with the reviewer. This is not a preliminary study. We have conducted a preliminary study already which were published in nutrients investigating this topic (Eckstein, M.L.; Brockfeld, A.; Haupt, S.; Schierbauer, J.R.; Zimmer, R.T.; Wachsmuth, N.; Zunner, B.; Zimmermann, P.; Obermayer-Pietsch, B.; Moser, O. Acute Metabolic Responses to Glucose and Fructose Supplementation in Healthy Individuals: A Double-Blind Randomized Crossover Placebo-Controlled Trial. Nutrients 2021, 13, 4095, doi:10.3390/nu13114095). Increasing the sample size would be the incorrect approach since our sample size estimation delivered sufficient power. In addition, we have conducted 224 study visits (14 for each participant) to avoid any kind of inter-variability between subjects hence we have conducted each visit related study set-up with the same participant. The power of our study was at 90% with a two-sided alpha. Other studies in the field conducted studies with half the sample size following a sufficient sample size estimation without the consideration of referring to it as a ‘preliminary trial’ (Baur, D.A.; Schroer, A.B.; Luden, N.D.; Womack, C.J.; Smyth, S.A.; Saunders, M.J. Glucose-fructose enhances performance versus isocaloric, but not moderate, glucose. Med. Sci. Sports Exerc. 2014, 46, 1778–1786, doi:10.1249/MSS.0000000000000284). It is not necessary and would be unethical to increase the sample size without any statistical backup. We would like to emphasize that we had three different types of exercise with four types of supplementation which was conducted by all 16 individuals of the study. Consequently, we believe the sample size estimation in addition to the sufficient power of the study in comparison to other studies in the field will remain endure and be sufficient for a publication in Nutrients.

  1. Material and Methods

Line 103 “…..of healthy individuals”.  It's not quite true because on line 111 the authors wrote ….BMI 18.0-29.9 . This is overweight, almost obese (BMI is from 30 and above). And  in line 115….. the range of 90–150 and 50– 95 mmHg, and 150/90 mmHg is stage I hypertension. Please,, explain it.

Thank you for you comment. We agree partly with you on this behalf but we understand your concern. Since we have recruited our participants from an athletic pool muscle mass could have been an influential factor that could have negatively influenced the BMI even though the participant was neither overweight or obese. However, as you can see in the result section we had a normal distributed sample size in BMI which was 22.6 ± 1.8 kg/m2 (BM 70.9 ± 10.8 kg, height 1.76 ± 0.08 m). Our included cohort is therefore a sufficient example of the german population within this age range since the average german man within this age group is has a BMI between 23.2 and 24.2 while the average BMI for women is 22.3. Our participant represent this group rather well since the numbers are in line. In addition, no participant was included with a BMI >25.

Regarding your second comment: This is the study inclusion criteria which is regularly influenced by white-coat hypertension (https://www.mayoclinic.org/diseases-conditions/high-blood-pressure/expert-answers/white-coat-hypertension/faq-20057792#:~:text=You%20could%20have%20white%20coat,pressure%20sometimes%20wear%20white%20coats.) Participants were screened by experienced study physicians; hence no participant was suffering under hypertension stage I. Blood pressure values that were measured and deemed implausible were repeated and if necessary checked with prior / external medical documentation for the participant.

Line 111- “Participants had to be metabolically healthy…  what do you mean the metabolically healthy, Please, explain it.

Thank you for your comment. This implies that no participant had any history of metabolic disease, was in remission or taking drugs/substances, performing diets to remain metabolically healthy. In addition blood glucose values were checked prior to inclusion of the study to ensure that the metabolic response to the consumption of carbohydrates was normal.

2.1. Eligibility criteria .   The criterion for inclusion or exclusion of participants from the study is not precisely described. Please, correct it.

Thank you for your comment. Due to the tabular description of the inclusion and exclusion criteria we have added these details in the supplemental material. We hope this is acceptable for the reviewer.

Page 3 Line 125-126

“The inclusion and exlcusion criteria can be found in the supplemental material (S1)”

Inclusion Criteria:

  • Informed consent obtained
  • Male or female aged 18–65 years (both inclusive)
  • Body mass index 18.0–29.9 kg/m2 (both inclusive)
  • Body Mass specific oxygen uptake >20 ml/min/Kg-1 (VO2max)
  • Sufficient lower- and upper body strength to conduct movements in leg press and chest press exercise (100% BM leg-press and 30%/75% BM chest-press)
  • Normal glucose tolerance (measured via overnight fasting blood glucose levels)

Exclusion Criteria:

  • Enrolment in other study
  • Known or suspected hypersensitivity to trial product(s) or related products
  • Receipt of any investigational medicinal product within 1 week prior to screening in this trial
  • Suffer from or history of a life-threatening disease (i.e. cancer judged not to be in full remission except basal cell skin cancer or squamous cell skin cancer), or clinically severe diseases that directly influence the study results, as judged by the Investigator. This does not prohibit the participation of patients taking medications that influences the metabolism (e.g. statin) or cardio-respiratory system (e.g. asthma spray) as long as the therapy is stable and is not adapted throughout the run of the trial. Furthermore, it does not excluded patients how have celiac disease (or similar diseases or allergies), as long as the disease is stable, and patients are able to stay on their specific (e.g.) gluten-free diet.
  • Participant with a heart rate <35 beats per minute (bpm) at screening (after resting for 5 min in supine position)
  • Supine blood pressure at screening (after resting for 5 min in supine position) outside the range of 90─150 mmHg for systolic or 50─95 mmHg for diastolic (excluding white-coat hypertension; therefore, if a repeated measurement on a second screening Visit shows values within the range, the participant can be included in the trial). This exclusion criterion also pertains to participants being on anti-hypertensives
  • Significant abnormal ECG at screening, as judged by the Investigator
  • Any chronic (metabolic) disorder or severe disease which, in the opinion of the Investigator might jeopardise participant’s safety or compliance with the protocol
  • History of multiple and/or severe allergies to drugs or foods or a history of severe anaphylactic reaction
  • Participant with mental incapacity or language barriers precluding adequate understanding or cooperation or who, in the opinion of their general practitioner or the Investigator, should not participate in the trial
  • Any condition that would interfere with trial participation or evaluation of results, as judged by the Investigator
  • Female of childbearing potential who is pregnant, breast-feeding or intend to become pregnant or is not using adequate contraceptive methods (adequate contraceptive measures include sterilisation, hormonal intrauterine devices, oral contraceptives, sexual abstinence or vasectomised partner).

2.2. Study design.

What was the sample selection?

Thank you for your comment. Participants were recruited via the University of Bayreuth, e-mail or newspapers. Potential participants were allowed to contact the study team and were given information prior to voluntarily offer to be involved for a screening visit and the additional 12 visits of the study

When was the study conducted?

Thank you for your comment. The study was conducted from November 2021 until April 2022.

How many trial visits were there?

Thank you for your comment. Including study visits we have conducted 224 trial visits. (13 visits for each participant)

This has been added in

Page 3 Line 111-113.

“The study was conducted between November 2021 and ended in April 2022. Overlal 224 trial visits were conducted.”

Where and how were participants selected for the study?

Thank you for your comment. Individuals that were interested in the study were allowed to contact the study team or the PI of the study directly to receive information about the study. Participants were deemed eligible for the study once they completed the screening visit and fulfilled all criteria shown in Table S1.

Did the participants receive any financial gratification?

Thank you for your comment. The participants did not receive any financial gratification. However, they have received detailed medical information and cardio-respiratory tests worth roughly 700€ during the course of the study.

Line 158 Participants also were asked about their physical activity via an international physical 158 activity questionnaire in short form (IPAQ-SF). There is no references, please add it.

Thank you for your comment. Apologies for not adding it in the first place. This has been done now

Line 168 Blood glucose and lactate values were collected via capillary measurements  from the earlobe at rest, after a 3-minute warm-up phase, every minute, at the end and after cool down and quantified afterwards.  How many times?

      Line 188….the participant was sent home and the visit was rescheduled. How and when were they re-enrolled in the study?

  1. Results

Please add age range of participants.

Line 257-259 One screened participant was not eligible to participate in the study because of a fructose hypersensitivity. No participant had to be  withdrawn or left the study prematurely.

But in Line 261 14 out of 16 participants were included for analysis

And in Line 261-4 One individual had an episode  of asymptomatic ventricular extrasystoles with a 4:1 ratio during the first study specific cycling test. This might trigger ventricular arrhythmias during the test as judged by the medical investigator and therefore he was excluded for further cycling exercises for safety  reasons.

Please explain it, how many participants took part in this study.

Thank you for your comment. In this study 16 individuals took part. This has also been described in the results section.

Page 7 Line 262 – 269

“The group of participants consisted of 16 healthy adults (9 females) with an mean ± SD age of 23.8 ± 1.6 years and a BMI of 22.6 ± 1.8 kg/m2 (BM 70.9 ± 10.8 kg, height 1.76 ± 0.08 m). Male participants had a relative VO2max of 52 ± 8 ml/kg/min, a peak power of 334 ± 25 W and a maximum heart rate of 190 ± 8 bpm at the screening CPX test. Female participants had a relative VO2max of 41 ± 5 ml/kg/min, a peak power of 236 ± 33 W and a maximum heart rate of 188 ± 8 bpm. One screened participant was not eligible to participate in the study because of a fructose hypersensitivity. No participant had to be withdrawn or left the study prematurely. Detailed study results can be seen in figure 2.“

  1. Discussion

In the discussion, the authors from line 358 to 380, almost one page, describe the results, this should be described in the 3rd results. This is not a discussion. Please correct it.

Thank you for your comment. We have shortened the discussion and hope this is in agreement with the reviewers suggestion.

Page 15 Line 361-451

Our findings deliver a lot of information that should be carefully considered by athletes, coaches and real-world practitioners. First and foremost, the results of our study are not entirely new, since previous studies found likely benefits of CHO supplementations in comparison to placebo during cycling [18,24]. Others omitted the effect of CHO on performance [19] and reviews cautiously recommended GluFru for moderate to- high-intensity exercise sessions [20]. Considering this, our study demonstrates clear evidence that pre-exercise CHO consumption, independent of composition does not have an ergogenic effect on performance in high-intensity anaerobic endurance exercise and moderate resistance exercise. The metabolic responses at rest seen in this study following the consumption of CHO have previously been investigated by our research group [17,25,26]. On this basis, this study was conducted, since the time until reaching peak blood glucose and peak lactate values are reached in healthy individuals roughly 30 minutes after consumption which could have had the implication of a higher performance in comparison to placebo. Previous studies conducting similar exercise tests on a cycle ergometer at roughly 85% of VO2max showed significant improvements in performance in half the sample size and solely male participants which cannot be considered as representative [27]. It might be considered that pre-exercise CHO consumption may lead to altered results between genders, since placebo was favored in women while in men it led to the 2nd best outcome with marginal differences to glucose during cycling. A previous review has highlighted an altered CHO response in women compared to men [28]. Women have shown a significantly decreased CHO oxidation in response to glucose [29]. Women generally oxidize less CHO and more lipids due to a higher presence of estrogens [30]. Female contraceptive use is associated with low estrogen levels and a reduction in insulin levels which all female participants in our study were consuming that could have had an impact on our findings [31].

Even though CHO supplementation does not lead to ergogenic effects on performance during high-intensity anaerobic endurance exercise and moderate resistance exercise, one should carefully consider that gender specific differences in response to glucose and fructose are evident that may demand future investigation. From this aspect our study is not without limitations. Since the included participants were young and healthy, yet without a specific training background of the tested types of exercise. It would be of interest how CHO supplementation would impact performance in well-trained cyclists or in well-resistance trained individuals. Due to the low amount of weight, the measured standard deviations were quite high and led to a large inter-individual variability that could have influenced levels of statistical significance in our study. In addition, we did not hand out food diaries over the entire course of the study since participants were asked to arrive fasted prior to each visit to avoid any kind of bias. Participants were encouraged to replenish glycogen storage after each visit which was ensured via a 48h break between visits.  However, for safety reasons and level of experience of the participants we were unable to increase the weights. Since the metabolic responses to the applied protocol at rest are well researched, our findings may deliver valuable information for male and female athletes consuming CHO drinks regularly prior to exercise [17,25,26].“

The weakest element of the study is too small a study population.

Thank you for your comment. As replied earlier to your comment, the sample size has been calculated a priori. No other study in the field investigating the effects of individualized carbohydrate supplementation schemes on different types of exercise in a cross-over setting had a larger sample size. Hence, we believe that this sample size is sufficient and deemed valid to underpin the findings of our study.

Reviewer 3 Report

The manuscript describes an experimental design that tested whether supplementation with glucose, fructose or a 50-50 mixture of the two sugars was able to influence both anaerobic and aerobic performance in a group of 16 healthy subjects (nine women).

The results obtained, analysed according to the parameters of a double-blind randomised placebo-controlled crossover trial, did not show any significant effects in the strength and endurance test aspects.

In my opinion, the lack of statistically significant differences could also be justified by what is now known in the literature on the use of glucose and/or fructose supplementation during the performance of endurance and/or strength activities. 

Furthermore, despite the fact that the authors report a correct procedure for determining the size (n=16) of the test subjects in order to obtain significant replies (Baur et al. using a paired t-test), I am rather dubious in considering that experiments conducted on 16 individuals (sometimes 14), are sufficient to obtain significant reply to statistical analysis when comparing, against placedo, three different experimental situations even if on the same individuals.

A final consideration concerns the homogeneity of the sample used. As the authors themselves report in the discussion section, working on such small numbers of experimental subjects, it would be preferable to select the study participants in such a way as to ensure greater homogeneity. For example, more truthful results could perhaps be obtained with athletes who perform the same speciality and who, presumably, implement a similar physical preparation programme.  

I hope that these thoughts may be considered useful by the authors of the manuscript for the future of their intriguing experimental project.

Author Response

Reviewer 3

The manuscript describes an experimental design that tested whether supplementation with glucose, fructose or a 50-50 mixture of the two sugars was able to influence both anaerobic and aerobic performance in a group of 16 healthy subjects (nine women).

The results obtained, analysed according to the parameters of a double-blind randomised placebo-controlled crossover trial, did not show any significant effects in the strength and endurance test aspects.

In my opinion, the lack of statistically significant differences could also be justified by what is now known in the literature on the use of glucose and/or fructose supplementation during the performance of endurance and/or strength activities. 

Furthermore, despite the fact that the authors report a correct procedure for determining the size (n=16) of the test subjects in order to obtain significant replies (Baur et al. using a paired t-test), I am rather dubious in considering that experiments conducted on 16 individuals (sometimes 14), are sufficient to obtain significant reply to statistical analysis when comparing, against placedo, three different experimental situations even if on the same individuals.

A final consideration concerns the homogeneity of the sample used. As the authors themselves report in the discussion section, working on such small numbers of experimental subjects, it would be preferable to select the study participants in such a way as to ensure greater homogeneity. For example, more truthful results could perhaps be obtained with athletes who perform the same speciality and who, presumably, implement a similar physical preparation programme.  

I hope that these thoughts may be considered useful by the authors of the manuscript for the future of their intriguing experimental project.

Thank you very much for your considerate thoughts and comments to our study and reading our manuscript in detail. We agree with you that a specific (well-trained) group of athletes may lead to different results. However, the participants in our study were mostly exercise science students that were not highly experienced in these types of exercise but however not entirely unfamiliar with the movements.

We do appreciate your input and will consider it for future projects of ours and again would like to thank you for your comments.

Round 2

Reviewer 1 Report

i accept in the present form